# Comprehensive Evaluation of Non-Standard Inflammatory and Metabolic Indices in Obesity: A Single-Center Retrospective Study

**DOI:** 10.3390/healthcare13222946

**Published:** 2025-11-17

**Authors:** Latife Merve Yildiz, Bayram Kızılkaya, Osman Cüre

**Affiliations:** 1Department of Family Medicine, Ordu University Faculty of Medicine, 52200 Ordu, Turkey; 2Department of Internal Medicine, Faculty of Medicine, Recep Tayyip Erdoğan University, 53200 Rize, Turkey; dr.bayram_kizilkaya@hotmail.com; 3Department of Rheumatology, Faculty of Medicine, Recep Tayyip Erdoğan University, 53200 Rize, Turkey; creosman61@gmail.com

**Keywords:** non-standard inflammatory indices, obesity, obesity-related inflammation, systemic immune-inflammation index (SII)

## Abstract

**Background:** Obesity is a major public health concern that predisposes individuals to metabolic and cardiovascular complications through chronic inflammation. This study aimed to evaluate the associations between non-standard inflammatory and metabolic indices [Systemic immune-inflammation index (SII), neutrophil/lymphocyte ratio (NLR), platelet/lymphocyte ratio (PLR), monocyte/lymphocyte ratio (MLR), C-reactive protein/albumin ratio (CRP/Alb), Fibrosis-4 (FIB-4), triglyceride/high-density lipoprotein (TG/HDL), and triglyceride/glucose (TG/Glucose)] and clinical variables in individuals with obesity. **Methods:** This single-center retrospective study included 298 adults with varying body mass index (BMI) categories followed at Recep Tayyip Erdoğan University Training and Research Hospital between February 2023 and February 2024. Demographic, clinical, and laboratory data were collected. Hematological (leukocyte, neutrophil, lymphocyte, monocyte, and platelet) and biochemical [glucose, triglyceride, HDL, albumin, CRP, aspartate aminotransferase (AST), and alanine aminotransferase (ALT)] parameters were analyzed, and derived indices (CRP/Alb, TG/Glucose, FIB-4, and SII) were calculated. Statistical analyses were performed using SPSS 25.0, and *p* < 0.05 was considered significant. **Results:** The mean age was 38.9 ± 11.5 years, and 54% were female. A positive correlation was found between BMI and CRP/Alb (r = 0.145, *p* = 0.012), whereas age showed a positive correlation with FIB-4 (r = 0.409, *p* < 0.001) and a negative correlation with SII (r = −0.117, *p* = 0.044). FIB-4 and SII were negatively correlated (r = −0.294, *p* < 0.001). TG/HDL and TG/Glucose were higher in men, whereas PLR was higher in women (*p* < 0.05). CRP/Alb was elevated in diabetes, and FIB-4 was higher in hypertension and hyperlipidemia. **Conclusions:** Non-invasive inflammatory and metabolic indices were significantly associated with obesity-related parameters. FIB-4, CRP/Alb, and TG/HDL may serve as complementary indicators of metabolic and fibrotic burden, reflecting the inflammatory–metabolic profile of individuals with obesity.

## 1. Introduction

Obesity is a major global public health concern. According to the World Health Organization (WHO), one in eight individuals worldwide is currently obese, accounting for nearly 890 million adults, while 2.5 billion are overweight [1]. In Türkiye, data from the 2022 TURKSTAT Health Survey indicate that 20.2% of adults are living with obesity, with higher rates among women (23.6%) than men (16.8%) [2].

Obesity is not merely a matter of excess body weight, it is strongly associated with type 2 diabetes, hypertension, cardiovascular diseases, non-alcoholic fatty liver disease (NAFLD), hepatic fibrosis, and several types of cancer [3]. Inflammation is widely recognized as a key mechanism linking obesity to these pathological conditions. In recent years, non-standard inflammatory indices such as systemic immune-inflammation index (SII), neutrophil/lymphocyte ratio (NLR), platelet/lymphocyte ratio (PLR), monocyte/lymphocyte ratio (MLR), C-reactive protein/albumin ratio (CRP/Alb) and Fibrosis-4 score (FIB-4) have gained increasing attention, particularly for the early detection of metabolic and hepatic disorders.

The significance of these non-standard inflammatory and metabolic indices has been consistently supported by recent studies. Elevated SII, NLR, and MLR levels have been associated with increased NAFLD risk, while systematic reviews and meta-analyses have highlighted NLR and PLR as reliable indicators of hepatic inflammation [4,5,6,7,8]. Moreover, recent evidence has identified NLR and the Systemic Inflammation Response Index (SIRI) as key determinants of liver fibrosis [9]. Collectively, these indices represent interconnected inflammatory and metabolic pathways associated with obesity, with SII, NLR, PLR, and MLR reflecting systemic immune activation, CRP/Alb indicating inflammation–nutrition balance, and FIB-4, TG/HDL, and TG/Glucose ratios capturing hepatic and metabolic dysfunction.

The aim of this study was to evaluate the associations of non-standard inflammatory and metabolic indices with demographic characteristics and clinical comorbidities in individuals with obesity, thereby providing a more comprehensive understanding of the metabolic and fibrotic risks associated with the condition and clarifying the role of inflammation in disease progression.

## 2. Methods

This single-center, retrospective, observational study was conducted at the Department of Internal Medicine, Recep Tayyip Erdoğan University Training and Research Hospital, between February 2023 and February 2024. Blood samples obtained from adult individuals with different body mass index (BMI) levels were evaluated. Data were retrieved from the hospital information system. Participants aged 18 years and older with complete demographic, clinical, and laboratory data were included in the study, while those with chronic inflammatory, autoimmune, infectious, or malignant diseases, as well as those with missing or incomplete records, were excluded.

Demographic characteristics (age, sex, marital status, and educational level), clinical diagnoses, treatments received, and laboratory parameters of the patients were recorded. Comorbidities such as diabetes mellitus, hypertension, and hyperlipidemia were confirmed through medical records and recorded as categorical variables (present/absent) for statistical analysis. The evaluated laboratory parameters included hematological indices [(leukocyte, neutrophil, lymphocyte, monocyte and platelet) (Mindray BC-6000 hematology analyzer, Mindray, Shenzhen, China)] and biochemical parameters [(glucose, triglyceride, HDL, albumin, CRP, aspartate aminotransferase (AST), alanine aminotransferase (ALT)), (Beckman Coulter AU5800 autoanalyzer, Beckman Coulter, Brea, CA, USA)]. From these parameters, the following indices were calculated: CRP/Alb ratio, TG/glucose ratio, FIB-4 score, and SII.

Calculation of Derived Indices: The derived inflammatory and metabolic indices were calculated using standard formulas. The neutrophil-to-lymphocyte ratio (NLR) was defined as neutrophil count divided by lymphocyte count, and the platelet-to-lymphocyte ratio (PLR) and monocyte-to-lymphocyte ratio (MLR) were calculated similarly. The systemic immune-inflammation index (SII) was computed as (platelet × neutrophil)/lymphocyte. The C-reactive protein to albumin ratio (CRP/Alb) was obtained by dividing CRP (mg/L) by albumin (g/dL). The FIB-4 index was calculated as (Age × AST)/[Platelet × √ALT]. The triglyceride-to-HDL (TG/HDL) ratio and triglyceride-to-glucose (TG/Glucose) ratio were determined by dividing serum triglyceride (mg/dL) by HDL-cholesterol (mg/dL) and fasting glucose (mg/dL), respectively.

Obesity is defined by the WHO as BMI ≥ 30 kg/m^2^. In clinical practice, this classification is further divided into subcategories accepted by the Centers for Disease Control and Prevention (CDC): Class I obesity (BMI 30.0–34.9 kg/m^2^), Class II obesity (BMI 35.0–39.9 kg/m^2^), and Class III obesity (BMI ≥ 40 kg/m^2^) [10,11].

The degree of hepatic steatosis was evaluated based on abdominal ultrasonography findings obtained from patient records. Steatosis was graded semi-quantitatively according to echogenicity, visualization of intrahepatic vessels, and diaphragm clarity, following conventional ultrasonographic criteria (Grade 0: none, Grade 1: mild, Grade 2: moderate, Grade 3: severe). Smoking and alcohol consumption status were also retrieved from patient records. Individuals who reported active cigarette use at the time of admission or within the previous 12 months were classified as “smokers.” Alcohol use was defined as the consumption of alcoholic beverages at least once per week. Participants who had never smoked or consumed alcohol, or who had ceased consumption more than 12 months prior to enrollment, were categorized as “non-smokers” and “non-users”, respectively.

## 3. Statistical Analysis

All statistical analyses were performed using IBM SPSS Statistics version 25.0 (SPSS Inc., Chicago, IL, USA). Continuous variables were expressed as mean ± standard deviation (SD) or median (interquartile range, IQR), while categorical variables were presented as frequencies (n) and percentages (%).

Spearman’s rank correlation analysis was conducted to assess relationships between continuous variables, as none of the data followed a normal distribution. For group comparisons, non-parametric tests were applied. The Mann–Whitney U test was used for comparisons between two groups, and the Kruskal–Wallis H test was used for comparisons among multiple groups. When significant differences were identified, Bonferroni-corrected post hoc analyses were performed. A *p*-value of <0.05 was considered statistically significant.

The study was conducted in accordance with the principles of the Declaration of Helsinki. Ethical approval was obtained from the **Recep Tayyip Erdoğan University Non-Interventional Clinical Research Ethics Committee** (approval number: **E-40465587-050.01.04-1200, dated 15 August 2024**). Due to the retrospective design, informed consent was waived; however, all data were anonymized and patient confidentiality was strictly preserved.

## 4. Results

A total of 298 participants with varying BMI categories were included in the study, and their demographic and clinical characteristics are summarized in Table 1.

Table 1 presents the frequency analysis results of the participants’ demographic and various clinical variables. According to this analysis, 29.2% (n = 87) of the participants were aged 18–30 years, 54.0% (n = 161) were female, 30.6% (n = 91) were classified as Class I obesity, 73.2% (n = 218) reported not smoking, 96.0% (n = 286) reported not consuming alcohol, 68.1% (n = 203) were married, and 33.9% (n = 101) were university graduates. Regarding chronic diseases, 6.0% (n = 18) of the participants had diabetes, 6.4% (n = 19) had hypertension, and 3.7% (n = 11) had hyperlipidemia.

Table 2 presents the minimum, maximum, mean, and standard deviation values of the participants’ scores for various indices.

Table 3 presents the correlation results between various variables and inflammatory and metabolic indices. The analysis revealed a positive correlation between age and FIB-4 (r = 0.409, *p* < 0.001), whereas age was negatively correlated with SII (r = −0.117, *p* = 0.044). A positive correlation was also observed between BMI and CRP/Alb (r = 0.145, *p* = 0.012). A significant negative correlation was observed between FIB-4 and SII (r = −0.294, *p* < 0.001). SII showed significant positive correlations with NLR (r = 0.763, *p* < 0.001), MLR (r = 0.303, *p* < 0.001), CRP/Alb (r = 0.236, *p* < 0.001) and PLR (r = 0.757, *p* < 0.001). TG/Glucose ratio demonstrated a strong positive correlation with TG/HDL (r = 0.911, *p* < 0.001). NLR was significantly correlated with MLR (r = 0.456, *p* < 0.001), CRP/Alb (r = 0.173, *p* = 0.003) and PLR (r = 0.320, *p* < 0.001). MLR also showed significant correlations with CRP/Alb (r = 0.168, *p* = 0.004) and PLR (r = 0.171, *p* = 0.003). These relationships are additionally illustrated as a heatmap in Figure 1, where red shades indicate positive correlations and blue shades represent negative correlations.

In addition to the correlation analyses illustrated in Figure 1, group comparisons were performed to further explore the associations between biomarkers and clinical or demographic variables. These comparative analyses are summarized in Table 4.

Table 4 presents the analysis results comparing biomarkers according to sex. TG/Glucose values were significantly higher in men than in women (**Z = 3.39, *p* < 0.001**). Similarly, TG/HDL ratios were also higher in men (**Z = 4.97, *p* < 0.001**). MLR levels were significantly elevated in males (**Z = 3.20, *p* = 0.001**), whereas PLR values were higher in females (**Z = −3.74, *p* < 0.001**). In contrast, no significant sex-based difference was observed in NLR levels (***p* = 0.947**). Additionally, CRP/Alb ratios were significantly higher in females compared to males (**Z = −2.01, *p* = 0.045**).

A significant association was found between BMI groups and TG/Glucose levels (H = 31.27, *p* < 0.001), which increased with higher BMI. Post hoc analysis showed that the normal group had significantly lower TG/Glucose values than the overweight (*p* = 0.0003), Class I obesity (*p* = 0.0001), and Class II obesity (*p* = 0.00002) groups. Similarly, TG/HDL ratios differed significantly among BMI groups (H = 36.91, *p* < 0.001) and progressively increased with obesity severity. The normal group exhibited lower TG/HDL values than the overweight (*p* = 0.00005), Class I (*p* = 0.00002), Class II (*p* < 0.0001), and Class III obesity (*p* = 0.00005) groups. In contrast, NLR levels varied modestly by BMI (*p* = 0.025), with higher values in the normal group compared to individuals with Class I obesity (*p* = 0.0187). Although PLR levels differed across BMI groups (*p* = 0.028), Bonferroni-corrected post hoc comparisons did not reveal significant pairwise differences (*p* > 0.05).

Furthermore, BMI groups were significantly associated with CRP/Alb levels (*p* < 0.001), which progressively increased with higher BMI. Post hoc analysis showed that the normal group had significantly lower CRP/Alb ratios than all obesity classes (*p* < 0.001), while individuals with Class III obesity exhibited higher CRP/Alb levels compared to the overweight (*p* = 0.0002) and Class I obesity (*p* = 0.0285) groups. In addition, CRP/Alb values were significantly higher in participants with diabetes (Z = −2.097, *p* = 0.036), whereas FIB-4 scores were elevated in those with hypertension (Z = −3.570, *p* < 0.001) and hyperlipidemia (Z = −2.490, *p* = 0.013).

## 5. Discussion

Obesity is a complex condition characterized by chronic low-grade inflammation, predisposing individuals to diverse metabolic and hepatic complications. In recent years, non-standard inflammatory and metabolic indices have been increasingly used to capture the inflammatory and fibrotic burden associated with obesity more comprehensively.

The positive correlation between age and FIB-4 in our study aligns with previous findings indicating that this index is influenced by age; fixed cut-off values may lead to false positives in older populations, suggesting the need for age-adjusted thresholds [12,13]. Conversely, the negative association between age and SII may reflect immunosenescence and inflammaging—while inflammatory stimuli increase with age, the decline in immune cell function may reduce SII levels [14,15]. Hence, interpreting SII as an isolated marker of inflammation in older individuals with obesity may be misleading.

The CRP/Alb ratio, reflecting both inflammation and nutritional status, has been proposed as a prognostic indicator in metabolic syndrome and liver disease [16]. Elevated CRP/Alb indicates chronic low-grade inflammation, combining increased hepatic CRP synthesis and decreased albumin production, both linked to insulin resistance and endothelial dysfunction [17,18]. The association between CRP/Alb and diabetes in our cohort may reflect regional inflammatory or nutritional patterns, consistent with prior studies [17,19]. Similarly, the positive relationship between BMI and CRP/Alb underscores systemic inflammation and metabolic risk in individuals with obesity. TG/HDL and TG/Glucose indices have also been shown to correlate strongly with insulin resistance and metabolic syndrome. Tani et al. [20] reported in a Japanese cohort that TG/HDL was more closely linked to visceral obesity and metabolic syndrome components in women than in men (AUC: 0.797 vs. 0.712; *p* < 0.0001). These findings are consistent with our results demonstrating associations between BMI, TG/HDL, TG/Glucose, and sex differences.

The negative association between FIB-4 and SII in our study is consistent with previous reports. A large NHANES analysis also demonstrated a significant inverse correlation between these indices, while showing that SII was positively associated with hepatic steatosis [21]. This suggests that fibrosis and inflammation in individuals with obesity may not progress in parallel. FIB-4 mainly reflects fibrosis burden through age and platelet count, whereas SII represents acute immune activation. Beyond its hepatic role, FIB-4 mirrors systemic metabolic dysregulation; chronic hepatic inflammation and early fibrotic remodeling contribute to insulin resistance, oxidative stress, and endothelial dysfunction—key pathways in cardiometabolic risk. Even at subclinical levels, elevated FIB-4 has been associated with greater carotid intima-media thickness, atherosclerotic load, and cardiovascular events [22,23]. These findings indicate that hepatic fibrogenesis parallels systemic inflammatory and metabolic stress, making FIB-4 a useful surrogate indicator of cardiometabolic dysfunction in obesity. Therefore, a combined assessment of these indices may better distinguish fibrotic and inflammatory activity.

Recent research has further clarified the role of SII as an indicator of systemic inflammation. Li et al. [9], found in NHANES data that SII, together with NLR and PLR, mediated obesity development, ranking second after NLR (%Mediate = 4.29%; *p* < 0.001). Similarly, Zhou et al. [24] observed a positive association between SII and obesity, and Liao et al. [25] reported that this relationship strengthened with higher BMI. Other hematologic indices, including NLR, PLR, and MLR, have shown variable results. Türkkan et al. [26] detected no significant differences between obese adolescents and controls, and Soesilo et al. [27] reported similar findings for MLR in overweight and obese women. Such discrepancies suggest that the behavior of these indices differs by population, age, and obesity phenotype. In our study, NLR, PLR, and MLR varied significantly across subgroups, likely reflecting population-specific inflammatory and metabolic responses. Overall, SII appears to capture both the magnitude and distribution of inflammation more effectively than conventional indices, supported by its strong correlations with other markers in our data.

The TG/HDL and TG/Glucose ratios have received increasing attention in evaluating obesity-related metabolic disorders. Yuan et al. [28] reported using NHANES data that the TG/HDL ratio outperformed TG and HDL alone in predicting metabolic dysfunction-associated steatotic liver disease (AUC = 0.732). Likewise, Poochanasri et al. [29] found it to be an independent predictor of 10-year cardiovascular risk, particularly in older adults. Huang et al. [30] further demonstrated that TG/Glucose was associated with obesity and hypertension, while derived indices such as the Triglyceride–Glucose (TyG) index, TyG-BMI, and TyG–Waist-to-Height Ratio (TyG-WHtR) strongly predicted cardiometabolic risk. These results suggest that the BMI–TG/Glucose relationship in our study reflects lipid–glucose imbalance and insulin resistance. Although the TG/Glucose ratio differs from the logarithmic TyG index, both capture similar lipid–glucose interactions. TG/Glucose and TG/HDL were strongly correlated but retained as they represent complementary aspects of metabolic dysfunction—TG/Glucose reflecting insulin resistance and TG/HDL indicating atherogenic dyslipidemia—providing a broader view of metabolic alterations across BMI categories.

Sex-related differences are another key consideration when interpreting inflammatory and metabolic indices. Prior studies have shown higher NLR, PLR, and SII levels in women than in men, likely due to immune and hormonal differences [31]. In contrast, men tend to have lower HDL and higher TG/HDL ratios, while women show higher HDL and platelet-based indices such as PLR [32]. In our study, TG/HDL values were higher in men and PLR higher in women, consistent with these reports, emphasizing the need to account for sex-based variations when interpreting clinical risk.

Inflammation is also central to the pathophysiology of diabetes. Beyond CRP/Alb, hematologic indices such as NLR, PLR, and SII have been linked to renal injury and cardiovascular outcomes in type 2 diabetes. Patro et al. [33] reported strong associations between SII, PLR, and proteinuria, while a meta-analysis of 56 studies confirmed NLR, PLR, SII, and SIRI as sensitive indicators of inflammation, with NLR most strongly related to cardiovascular mortality and renal failure [34]. Similarly, the TyG index has been independently associated with albuminuria and chronic kidney disease [35]. In our cohort, only CRP/Alb showed a significant association with diabetes, possibly due to population differences or the limited number of diabetic participants. These findings support the link between CRP/Alb and diabetes and suggest that hematologic and metabolic indices may serve as complementary tools for early detection and risk stratification of diabetic complications.

In the literature, hypertension has been consistently linked to liver fibrosis, and several studies have shown that elevated blood pressure increases fibrosis risk in individuals with MAFLD, with non-invasive indices such as FIB-4 reflecting this relationship [36,37,38]. Classical inflammatory indices also provide insights into hypertension; NLR and PLR are often elevated and predict cardiovascular risk [39], while SII is recognized as a strong marker of systemic inflammation, correlating with cardiovascular mortality [40]. The CRP/Alb ratio has been associated with inflammation and endothelial dysfunction, particularly in resistant hypertension [41]. Recent studies have also identified the TyG index and its derivatives as independent predictors of all-cause and cardiovascular mortality in hypertensive populations [42]. In this context, our finding that hypertension correlated only with FIB-4 supports the idea that hepatic fibrosis may serve as a key indicator of cardiometabolic risk in hypertensive individuals. Nonetheless, discrepancies with previous findings regarding other indices (NLR, PLR, SII, CRP/Alb, TyG) may stem from population differences or the limited number of hypertensive participants in our cohort.

In our study, participants with hyperlipidemia exhibited significantly higher FIB-4 scores, emphasizing the association between dyslipidemia and hepatic fibrosis. Hyperlipidemia promotes hepatocyte injury through free fatty acid accumulation, oxidative stress, and lipotoxicity, accelerating fibrotic processes. Recent studies have confirmed FIB-4 as a valuable non-invasive tool for predicting fibrosis risk in dyslipidemic individuals, especially within NAFLD and MAFLD groups [43]. Although no significant differences were observed for other indices, previous research has linked NLR, PLR, and SII with atherosclerosis and dyslipidemia, reflecting the role of inflammation in cardiometabolic risk [44]. Moreover, the TyG index has been associated with both lipid abnormalities and hepatic fibrosis via insulin resistance [45]. Collectively, these findings suggest that fibrosis burden may represent a dominant feature of dyslipidemia and that integrating multiple non-invasive indices could enhance risk stratification in individuals with obesity.

## 6. Limitations

This study has several limitations. First, due to its cross-sectional and retrospective design, only correlations between variables could be evaluated, and causal relationships cannot be inferred. Second, as it was conducted in a single-center Turkish cohort, potential ethnic or regional differences may limit the generalizability of the findings. Third, the relatively small number of participants with comorbidities such as diabetes, hypertension, or hyperlipidemia may have reduced the statistical power of subgroup analyses. In addition, detailed information on medication use (e.g., statins, antihypertensives, antidiabetic, or anti-inflammatory drugs) was not available for all participants, and their potential confounding effects on lipid and inflammatory markers could not be fully excluded. The lack of a healthy control group and the absence of multivariable regression or ROC analyses further limit the interpretation of independent and predictive relationships. Finally, liver fibrosis was assessed using a non-invasive index (FIB-4) without imaging confirmation, which may restrict clinical validation. Future prospective, multicenter studies with comprehensive medication data and adjusted analyses are warranted to validate these findings.

## 7. Conclusions

This study demonstrated that non-invasive inflammatory and metabolic indices, including FIB-4, CRP/Alb, TG/HDL, and TG/Glucose ratios, were significantly associated with clinical and demographic variables in obese individuals. Specifically, FIB-4 was related to hypertension and hyperlipidemia, CRP/Alb to diabetes, and TG/HDL and TG/Glucose to BMI, reflecting their complementary value in assessing both fibrotic and metabolic components of obesity. These findings highlight the potential clinical utility of such easily accessible biomarkers for comprehensive risk stratification and metabolic monitoring in obesity management. Future large-scale, prospective studies are warranted to validate these associations and to establish standardized reference thresholds for clinical practice.

## Figures and Tables

**Figure 1 healthcare-13-02946-f001:**
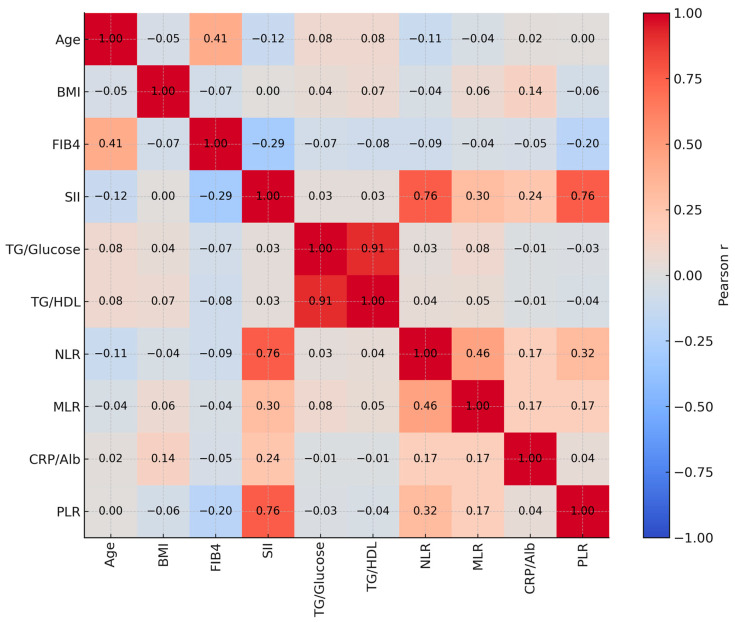
Correlation heatmap of clinical and inflammatory indices.

**Table 1 healthcare-13-02946-t001:** Frequency distribution of demographic variables of the participants (n = 298).

Variables	n or Median (Min−Max)	% or Mean ± SD
**Age (Years)**	38.50 (18.0–68.0)	38.92 ± 11.48
**Age group**		
18–30 years	87	29.2
31–40 years	79	26.5
41–50 years	77	25.8
>50 years	55	18.5
**Sex**		
Female	161	54.0
Male	137	46.0
**Height (cm)**	167.0 (142.0–190.0)	167.57 ± 9.48
**Weight (kg)**	90.0 (44.0–170.0)	92.80 ± 21.41
**Waist circumference (cm)**	106.0 (30.0–149.0)	106.07 ± 14.77
**BMI**	32.98 (17.19–57.07)	32.98 ± 6.51
**BMI group**		
Normal/Underweight	50	16.8
Overweight	55	18.5
Class I obesity	91	30.6
Class II obesity	53	17.8
Class III obesity	48	16.3
**Smoking status**		
No	218	73.2
Yes	80	26.8
**Alcohol consumption**		
No	286	96.0
Yes	12	4.0
**Marital status**		
Single/Widowed	95	31.9
Married	203	68.1
**Educational level**		
Primary school	40	13.4
Secondary school	39	13.1
High school	62	20.8
University	101	33.9
Postgraduate	56	18.8
**Diabetes**		
No	280	94.0
Yes	18	6.0
**Hypertension**		
No	279	93.6
Yes	19	6.4
**Hyperlipidemia**		
No	287	96.3
Yes	11	3.7

**Table 2 healthcare-13-02946-t002:** Descriptive statistics of participants’ scores for various indices (n = 298).

Variables	Min.	Max	Mean	SD
**FIB-4 Score**	0.11	7.46	0.71	0.56
**SII**	55.73	2964.11	521.95	313.81
**TG/Glucose**	0.00	7.74	1.51	0.96
**TG/HDL**	0.35	16.58	3.14	2.47
**NLR**	0.37	7.51	1.88	0.85
**MLR**	0.04	0.85	0.19	0.08
**CRP/Alb**	0.00	29.76	1.07	2.01
**PLR**	0.01	1.14	0.12	0.07

**Table 3 healthcare-13-02946-t003:** Correlation results between various variables and indices (n = 298).

		1	2	3	4	5	6	7	8	9	10
1-Age (Years)	r	1									
	*p*										
2-BMI	r	−0.050	1								
	*p*	0.392									
3-FIB-4 Score	r	0.409 **	−0.067	1							
	*p*	**<0.001**	0.247								
4-SII	r	−0.117 *	0.002	−0.294 **	1						
	*p*	**0.044**	0.977	**<0.001**							
5-TG/Glucose	r	0.078	0.044	−0.069	0.034	1					
	*p*	0.179	0.454	0.237	0.561						
6-TG/HDL	r	0.079	0.072	−0.079	0.034	0.911 **	1				
	*p*	0.175	0.214	0.177	0.559	**<0.001**					
7-NLR	r	−0.106	−0.044	−0.086	0.763 **	0.031	0.043	1			
	*p*	0.068	0.446	0.138	**<0.001**	0.590	0.461				
8-MLR	r	−0.045	0.056	−0.039	0.303 **	0.081	0.049	0.456 **	1		
	*p*	0.441	0.334	0.502	**<0.001**	0.163	0.398	**<0.001**			
9-CRP/Alb	r	0.02	0.145 *	−0.053	0.236 **	−0.015	−0.007	0.173 **	0.168 **	1	
	*p*	0.735	**0.012**	0.361	**<0.001**	0.800	0.907	**0.003**	**0.004**		
10-PLR	r	0.002	−0.062	−0.203 **	0.757 **	−0.033	−0.037	0.320 **	0.171 **	0.045	1
	*p*	0.971	0.287	**<0.001**	**<0.001**	0.573	0.528	**<0.001**	**0.003**	0.441	

* Correlation is significant at the 0.05 level (Spearman correlation test), ** Correlation is significant at the 0.01 level (Spearman correlation test). (The bold values indicate statistically significant results).

**Table 4 healthcare-13-02946-t004:** Comparison of indices with various variables (n = 298).

Variables	n	FIB-4 Score	SII	TG/Glucose	TG/HDL	NLR	MLR	CRP/Alb	PLR
Degree of steatosis		M(IQR)	M(IQR)	M(IQR)	M(IQR)	M(IQR)	M(IQR)	M(IQR)	M(IQR)
0	52	0.65 (0.45)	413.05 (283.66)	1.16 (0.72)	2.29 (2.1)	1.51 (0.84)	0.17 (0.05)	0.64 (0.95)	0.16 (0.05)
1	95	0.64 (0.35)	469.56 (278.57)	1.17 (1.06)	2.13 (2.46)	1.71 (0.61)	0.18 (0.06)	0.53 (0.76)	0.12 (0.5)
2	110	0.61 (0.4)	455.24 (336)	1.23 (0.97)	2.46 (2.31)	1.81 (0.98)	0.18 (0.07)	0.83 (0.90)	0.11 (0.05)
3	41	0.56 (0.42)	435.71 (294.84)	1.36 (1.11)	2.86 (3.57)	1.63 (1)	0.19 (0.09)	0.6 (0.7)	0.12 (0.05)
*p*=		0.915	0.723	0.564	0.610	0.459	0.333	0.071	0.798
**Sex**		**M(IQR)**	**M(IQR)**	**M(IQR)**	**M(IQR)**	**M(IQR)**	**M(IQR)**	**M(IQR)**	**M(IQR)**
Female	161	0.57 (0.4)	466.85 (349.35)	1.20 (0.8)	2.01 (1.66)	1.71 (0.9)	0.17 (0.06)	0.77 (1.0)	0.12 (0.05)
Male	137	0.64 (0.41)	433.56 (273.16)	1.45 (1.14)	3.32 (3.2)	1.70 (0.8)	0.19 (0.07)	0.58 (0.7)	0.11 (0.04)
Z=		1.35	−1.41	3.39	4.97	0.07	3.20	−2.01	−3.74
*p*=		0.179	0.158	**<0.001**	**<0.001**	0.947	**0.010**	**0.045**	**<0.001**
**BMI Group**		**M(IQR)**	**M(IQR)**	**M(IQR)**	**M(IQR)**	**M(IQR)**	**M(IQR)**	**M(IQR)**	**M(IQR)**
(1) Normal/Underweight	50	0.59 (0.37)	475.03 (295.16)	0.88 (0.44)	1.35 (1.10)	1.93 (0.91)	0.18 (0.1)	0.22 (0.41)	0.12 (0.04)
(2) Overweight	55	0.67 (0.39)	454.03 (265.99)	1.47 (1.17)	2.42 (3.52)	1.71 (0.70)	0.17 (0.09)	0.44 (0.57)	0.12 (0.04)
(3) Class I obesity	91	0.65 (0.42)	403.96 (304.40)	1.34 (1.03)	2.73 (2.76)	1.59 (0.81)	0.18 (0.07)	0.8 (0.65)	0.10 (0.04)
(4) Class II obesity	53	0.55 (0.40)	428.99 (287.50)	1.49 (0.93)	3.06 (1.89)	1.62 (0.94)	0.19 (0.05)	0.75 (0.88)	0.11 (0.04)
(5) Class III obesity	48	0.55 (0.41)	533.70 (360.82)	1.22 (0.61)	2.63 (1.67)	1.88 (0.72)	0.18 (0.04)	1.15 (1.01)	0.12 (0.035)
*p*=		0.374	0.089	**<0.001**	**<0.001**	**0.025**	0.967	**<0.001**	**0.028**
Post Hoc=		-	-	1 < 2.3.4.5	1 < 2.3.4.5	1 < 3	-	1 < 2.3.4.5	-
**Smoking status**		**M(IQR)**	**M(IQR)**	**M(IQR)**	**M(IQR)**	**M(IQR)**	**M(IQR)**	**M(IQR)**	**M(IQR)**
No	218	0.64 (0.44)	448.76 (296.99)	1.22 (0.99)	2.177 (2.297)	1.70 (0.84)	0.17 (0.07)	0.66 (0.08)	0.11 (0.05)
Yes	80	0.61 (0.35)	472.75 (346.35)	1.303 (1.049)	2.854 (2.767)	1.87 (0.81)	0.19 (0.05)	0.67 (0.91)	0.11 (0.04)
Z=		−0.72	0.54	0.63	+1.77	1.37	1.52	0.74	−1.78
*p*=		0.475	0.592	0.530	0.077	0.171	0.130	0.458	0.075
**Alcohol consumption**		**M(IQR)**	**M(IQR)**	**M(IQR)**	**M(IQR)**	**M(IQR)**	**M(IQR)**	**M(IQR)**	**M(IQR)**
No	286	0.63 (0.42)	451.14 (310.58)	1.23 (1.03)	2.38 (2.44)	1.72 (0.87)	0.18 (0.07)	0.67 (0.87)	0.12 (0.05)
Yes	12	0.53 (0.28)	415.01 (213.25)	1.18 (0.68)	2.48 (3.07)	1.66 (0.52)	0.19 (0.11)	0.91 (1.13)	0.09 (0.05)
Z=		−0.701	−0.530	−0.289	−0.033	−0.393	−0.147	−0.604	−1.125
*p*=		0.483	0.596	0.773	0.974	0.694	0.883	0.546	0.261
**Diabetes**		**M(IQR)**	**M(IQR)**	**M(IQR)**	**M(IQR)**	**M(IQR)**	**M(IQR)**	**M(IQR)**	**M(IQR)**
No	280	0.62 (0.42)	450.28 (308.87)	1.22 (0.98)	2.36 (2.39)	1.72 (0.84)	0.18 (0.07)	0.65 (0.79)	0.12 (0.05)
Yes	18	0.66 (0.38)	448.27 (322.72)	1.6 (1.35)	3.2 (4.77)	1.61 (1.06)	0.17 (0.06)	1.21 (0.93)	0.11 (0.07)
Z=		−1.273	−0.147	−1.126	−1.104	−0.595	−0.937	−2.097	−0.322
*p*=		0.203	0.883	0.260	0.269	0.552	0.349	**0.036**	0.748
**Hypertension**		**M(IQR)**	**M(IQR)**	**M(IQR)**	**M(IQR)**	**M(IQR)**	**M(IQR)**	**M(IQR)**	**M(IQR)**
No	279	0.60 (0.4)	450.88 (311.18)	1.23 (1.04)	2.39 (2.5)	1.72 (0.86)	0.19 (0.07)	0.68 (0.89)	0.12 (0.05)
Yes	19	0.95 (0.48)	384.57 (308.03)	1.23 (0.93)	2.41 (1.97)	1.5 (0.76)	0.17 (0.1)	0.64 (0.48)	0.11 (0.05)
Z=		−3.570	−0.769	−0.256	−0.348	−0.659	−0.802	−0.502	−1.198
*p*=		**<0.001**	0.442	0.798	0.728	0.510	0.423	0.616	0.231
**Hyperlipidemia**		**M(IQR)**	**M(IQR)**	**M(IQR)**	**M(IQR)**	**M(IQR)**	**M(IQR)**	**M(IQR)**	**M(IQR)**
No	287	0.61 (0.41)	449.69 (310.85)	1.22 (1.01)	2.35 (2.37)	1.71 (0.86)	0.18 (0.07)	0.67 (0.89)	0.12 (0.05)
Yes	11	0.90 (0.43)	498.86 (286.32)	1.84 (1.32)	3.9 (4.13)	1.85 (1.51)	0.17 (0.13)	0.74 (0.52)	0.11 (0.07)
Z=		−2.490	−0.123	−1.200	−1.639	−0.258	−0.111	−0.206	−0.358
*p*=		**0.013**	0.902	0.230	0.101	0.796	0.912	0.836	0.720

Data are presented as median (IQR). Comparisons between two groups were performed using the Mann–Whitney U test, and comparisons among multiple groups were assessed with the Kruskal–Wallis test followed by Bonferroni-corrected post hoc analysis when appropriate. A *p*-value of < 0.05 was considered statistically significant. SD: Std. Deviation, M: Median, IQR: Interquartile Range.

## Data Availability

The dataset generated and analyzed during the current study is openly available in Zenodo at https://doi.org/10.5281/zenodo.17442195 (accessed on 25 October 2025).

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
