# Peer review of "Comprehensive Evaluation of Non-Standard Inflammatory and Metabolic Indices in Obesity: A Single-Center Retrospective Study"

_healthcare, 2025, doi:10.3390/healthcare13222946_

Round 1
Reviewer 1 Report
Comments and Suggestions for Authors
Please see file attached.

English language is accepted with minor typos related to statistical report.
Author Response
The corresponding revisions have been provided in the attached files for your kind review.

Reviewer 2 Report
Comments and Suggestions for Authors
The study presents an extensive evaluation of novel, non-invasive biomarkers in obese individuals, effectively integrating its findings with existing literature through clear and biologically plausible explanations. It thoughtfully addresses sex-related differences and underscores the clinical relevance of these biomarkers, supporting their potential utility in obesity-related risk assessment.
However, prior to consideration for publication, several important limitations require attention and resolution.
Introduction:
- Provides a solid background but lacks a clear rationale for selecting the specific biomarkers and does not detail the study design.
- Emphasizes obesity prevalence statistics more than addressing research gaps or formulating an explicit hypothesis.
- Omits discussion of potential confounding factors and does not clarify the generalizability of findings beyond the Turkish population.
Methods:
- The single-center design limits the applicability of results to broader populations or different clinical settings.
- There is insufficient detail on how comorbidities were accounted for in the analysis.
- Important exclusion criteria, such as chronic inflammatory or autoimmune diseases, are not specified.
- No power calculation or sample size justification is provided, raising concerns about statistical validity.
Results:
- The cross-sectional nature restricts interpretations to associations, without establishing causality.
- Key comorbidities such as diabetes and hypertension are underrepresented, reducing the power of subgroup analyses.
- The lengthy data collection period (2018–2024) introduces potential variability that is not addressed.
- Multivariate analyses adjusting for confounders like age, sex, or medication use are absent.
- Small subgroup sizes limit the reliability of some comparisons.
- Lack of longitudinal follow-up impedes understanding of biomarker dynamics over time.
Discussion:
- Some conclusions overreach the limitations inherent to cross-sectional data, implying causality or predictive value without sufficient evidence.
- Inconsistent findings, such as the association of CRP/Alb with diabetes, are not adequately explored or contextualized.
- Certain biomarkers (e.g., NLR, PLR, MLR) receive minimal discussion despite being part of the study.
- There is limited reflection on participant demographics and their impact on the generalizability of results
Author Response

(The authors gave the same response as above.)

Reviewer 3 Report
Comments and Suggestions for Authors
General comment
The main interest of this study lies in describing the concentrations of a wide range of relatively novel inflammation-related indices in patients with obesity. While the number of cases is sufficient, the authors limit themselves to proposing the usefulness of these indices as biomarkers based solely on simple correlations, which is misleading. Currently, there is a tendency to use the term biomarker too loosely. A true biomarker is not merely any parameter altered in a disease; it must reliably distinguish between different clinical situations, whether for diagnosis, prognosis, or patient outcomes. The authors have not evaluated any of these aspects. Their abstract concludes that “These findings support the broader clinical use of non-invasive inflammatory markers in obesity management.” Unfortunately, the presented results do not justify this claim. Considering these points, I cannot recommend publication of this article in Healthcare. However, I suggest that the authors consider preparing a substantially reduced version for submission to another journal, addressing the concerns raised by this and other reviewers.
Specific comments
1) Please use “patient with obesity” instead of “obese patient” throughout your manuscript, as person-first language is preferred to avoid stigma and emphasize the individual rather than the condition.
2) Define terms such as SII, NLR, PLR, MLR, CRP/Alb ratio, and FIB-4, and all the other abbreviations, including CRP, AST, etc., the first time you use them.
3) Many studies indicate that severe or morbid obesity (BMI ≥40, or ≥35 with comorbidities) is associated with specific metabolic characteristics and clinical features. If the number of cases allows, it would be interesting to include a table or figure comparing your biomarkers in patients with moderate versus severe obesity.
4) Some of these scores may be unfamiliar to many readers and should be described in detail in the Methods section.
5) I am a bit confused. You state that you studied 298 patients with obesity, but Table 1 shows that 87 of them were normal weight, underweight, or overweight. This means that only 211 patients truly had obesity. The 87 patients without obesity should be used as a control group to assess differences in the various parameters, but they should not be included in the “patients with obesity” group.
Author Response

(The authors gave the same response as above.)

Round 2
Reviewer 1 Report
Comments and Suggestions for Authors
Despite limitation of study however, it can be accepted in current form.
Comments on the Quality of English LanguageDespite limitation of study however, it can be accepted in current form.
Author Response
Comments 1: The English could be improved to more clearly express the research. Despite limitation of study however, it can be accepted in current form.
Responce 1: We sincerely thank the reviewer for the positive feedback.
As suggested, minor language improvements have been made to enhance clarity and readability, while keeping the scientific content unchanged.
Reviewer 2 Report
Comments and Suggestions for Authors
This revised version of the manuscript could be considered for acceptance.
Author Response
Comments 1: This revised version of the manuscript could be considered for acceptance.
Responce 1: We sincerely thank the reviewer for the positive and encouraging evaluation. We appreciate the constructive review and are pleased that the manuscript is considered acceptable in its current form.